# The Combined Effect of Nitrogen Treatment and Weather Conditions on Wheat Protein-Starch Interaction and Dough Quality

**Indrek Keres [1,\*], Maarika Alaru [1], Reine Koppel [2], Illimar Altosaar [3], Tiina Tosens [1] and Evelin Loit [1]**

1 Institute of Agricultural and Environmental Sciences, Estonian University of Life Sciences, Fr.R. Kreutzwaldi 5, 51006 Tartu, Estonia; maarika.alaru@emu.ee (M.A.); tiina.tosens@emu.ee (T.T.); evelin.loit@emu.ee (E.L.)
2 Estonian Crop Research Institute, J. Aamisepa 1, 48309 Jõgeva, Estonia; reine.koppel@etki.ee
3 Department of Biochemistry, Microbiology and Immunology, University of Ottawa, 75 Laurier Ave. E, Ottawa, ON K1N 6N5, Canada; illimar@proteinseasy.com
\* Correspondence: indrek.keres@emu.ee

**Abstract:** The objective of this field crop study was to compare the effect of organic (cattle manure, off-season cover crop) and mineral N ($NH_4NO_3$; 0, 50, 100° 150 kg N ha$^{-1}$) fertilizers on (i) gluten-starch interaction, and (ii) rheological properties of winter wheat dough. Data were collected from the long-term field experiment located in the Baltic Sea region (58°22′ N, 26°40′ E) in years 2013–2017. The amount of minuppueral N 150 kg ha$^{-1}$ applied in two parts before flowering ensured higher gluten content (31 ± 3.3%) and dough quality (81 ± 7.4 mm) due to more positive interactions between gluten proteins and starch granules. The quality of dough was more variable in organic treatments (ranged up to 33%) because the availability of organic N was more variable and sensitivity to the weather conditions was higher. The mean variability of different dough properties over trial years under organic treatments was 1.4–2.0 times higher than in the treatment with 150 kg N ha$^{-1}$.

**Keywords:** starch; gluten; nitrogen; mineral; organic; dough rheology

## 1. Introduction

Bread wheat (*Triticum aestivum* L.) is one of the most crucial crops for human sustenance and wellbeing. Unlike other grain flours, wheat flour can generate dough exhibiting a unique three-dimensional structure and viscoelasticity with added water [1]. Wheat dough is a complex mixture, where proteins link together to form the continuous reticular skeleton and in which starch granules act as filling components [2]. High- and low-molecular–weight glutenin subunits have been demonstrated as determinant factors for dough formation and quality [3]. The rheological properties of wheat dough are not only determined by proteins but also by other flour components and their interactions [4]. Approximately 70% of grain and 75% of flour weight is composed of starch. Further to its pivotal role in grain quality and dough functionality through its internal structure and physicochemical properties, it also contributes via starch-gluten interactions [5].

Wheat starch granules have been reported to have a trimodal size distribution [6,7]. The large A-granules (generally larger than 10 μm in diameter) are formed first in developing endosperm, whereas the small B-granules (smaller than 10 μm in diameter) are formed later during kernel development. The formation of very small C-type granules (less than 5 μm) is initiated very late in grain filling [8,9]. Starch granule particle size has been reported to affect dough rheological properties, wherein smaller granules increased dough's elastic characteristics [2,10]. The large A- and smaller B-type starch granules have significant differences in their physical and chemical properties [11,12]. The granule size distribution of wheat starch affects its functionality [13,14], resulting in identifiable quality levels of many final baking, pasta and other industrial products.

While several studies have reported the influence of either gluten or starch alone on the rheological properties of wheat dough quality [15,16], the protein and starch interaction effect on wheat dough properties has been seldom studied because of its complicated multifaceted nature [17,18], thus forming a major gap in knowledge important to the baking industry.

The nitrogenous nutrition of cereals (fertilizing with organic or mineral N fertilizer) is crucial for the development of grain yield, as well as the rheological properties of wheat dough. As organic production becomes more widespread [19], and farmers are urged to decrease mineral nitrogen input, it is important to clarify the extent to which organic fertilization affects gluten accumulation and starch-granule formation during grain filling, whilst the interaction of gluten and starch granules on the dough properties of cereals grown in organic or conventional systems invites further exploration. Earlier, we demonstrated that in addition to the fertilizing regime, local weather conditions significantly influence N availability and the protein content of grains [20].

The objective of this study was to compare the effect of different nitrogen fertilizers (mineral or organic N) and post-hibernation weather conditions on (i) gluten-starch interaction, and (ii) stability of dough rheological properties of winter wheat flours.

**Hypotheses 1 (H1).** *Wheat grown under a mineral nitrogen background has dough of higher quality.*

**Hypotheses 2 (H2).** *The quality of the dough is more variable in organic treatments.*

## 2. Materials and Methods

Grain samples of winter wheat cv Fredis were harvested from a long-term crop rotation experiment, established in 2008 at the Estonian University of Life Sciences (58°22′ N, 26°40′ E). Field experimental conditions followed the previously described methodology. The experimental design, setup, chemical analyses, weather conditions (temperatures and precipitation during growing period) and measurement of starch-granule distribution are detailed in Alaru et al. [21], Keres et al. [20,22].

### 2.1. Experimental Setup

The rotation consisted of five field crops that followed each other in this order: barley (*Hordeum vulgare* L.) with undersown red clover, red clover (*Trifolium pratense* L.), winter wheat (*Triticum aestivum* L.), pea (*Pisum sativum* L.), and potato (*Solanum tuberosum* L.). All crops were grown every year. The field experiment had a systematic block design with four replicates that included the treatments of organic and mineral fertilization. The treatments of mineral fertilization in the conventional system was further divided into four subplots (10 × 6 m) corresponding to the mineral fertilizer (ammonium nitrate) rates of 0, 50, 100 and 150 kg N ha$^{-1}$. Three organic fertilization subplots were labelled Org 0, Org I and Org II. The data regarding winter wheat from this second rotation period were gathered during crop years 2013–2017.

Zero nitrogen (N0) was the control treatment for the conventional system, without mineral fertilizers, but with pesticides. The conventional treatment N50 had the mineral fertilizer ammonium nitrate (34,4N) applied in early spring at the tillering phase of winter wheat, while treatments for N100 and N150 had N fertilizer applied twice: (1) N100—50 + 50 kg N ha$^{-1}$ at tillering and booting stage BBCH47, respectively; (2) data for treatment of N150 were 100 + 50 kg N ha$^{-1}$. The three conventional treatments (N50, N100 and N150) had phosphorous (P) and potassium (K) fertilizers applied at sowing at the rate of 25 kg P and 95 kg K ha$^{-1}$ (Yara Mila Cropcare 3–11–24 was used). Amounts of P and K were similar in all treatments.

The organic treatment Org 0 was used as a control, with symbiotically fixed atmospheric N$_2$ by red clover and pea in the rotation as the only source of N, ploughed into the soil with the above-ground biomass (i.e., twice during the crop cycle period; red clover being pre-crop for winter wheat). In the organic treatment of Org I, in addition to legumes in the rotation, cover crops were used as green manure in winter (after winter wheat,

potato and pea). Cover crops were ploughed into the soil as soon as possible after the snow melted in April. In the organic treatment Org II the cattle manure was added in early spring before winter wheat re-growth at a rate of 10 t ha$^{-1}$, i.e., $50 \pm 4$ kg N ha$^{-1}$. Winter wheat was harvested with a Sampo combine on 12 August 2013, 4 August 2014, 12 August 2015, 26 July 2016 and 28 August 2017 (moisture content of kernels ranged from 20 to 28 percent).

### 2.2. Chemical Analyses

The total nitrogen (Ntot) content of the grain samples was determined by the dry combustion method on a varioMAX CNS elemental analyzer (ELEMENTAR, Hanau, Germany). Wet gluten content (WGC) was determined according to ISO standard 5531 (ISO 5531) by a Glutomatic 2100 apparatus (Perten Instruments AB, Huddinge, Sweden). Gluten index was measured with Perten's apparatus (ICC 155; Glutomatic 2100, Centrifuge 2015; Perten Instruments (now PerkinElmer, Waltham, MA, USA).

### 2.3. Determination of Dough Properties

Water absorption of flour and dough mixing properties were examined by the Brabender Farinograph–TS Version 2.1.0 (Brabender GmbH & Co, Duisburg, Germany) using the Brabender ICC BIPEA 50 method. Analyses were performed in accordance with ISO 5530-1 standard. The principle of farinograph operation is based on the resistance of dough to kneading. Farinograph curves show the time of formation, i.e., development of the dough, time of stability, and the degree of softening of the dough (after 10 and 12 min). Dough development time (DDT; min) defines the duration from the start of mixing to the point of maximum viscosity, while dough stability (S) is the time (min) when top of the farinograph crosses the 500 Brabender Units (BU) line to the point when it drops below it. The degree of softening (DS; FE) is the difference in height between the centre of the graph at maximum resistance to mixing and the centre of the graph at a point 10 or 12 min later. Dough quality number (DQN) is the length (mm) from the water point to a point 30 FE below the centre line of greatest consistency along the time axis. Low DQN indicates weak flour that weakens early and quickly while high DQN indicates strong flour that weakens late and slowly [23].

The alveograph (Chopin Technologies, Villeneuve la Garenne, France) was used to measure wheat viscoelastic properties (AACC Approved Method 54–30.02, ICC 121, ISO 27971:2015) [24]. The alveograph measures the main parameters of dough response to biaxial extension by inflating it with air, ie., the pressure (expressed by parameter W) generated inside the dough bubble and the deformation of the dough piece until it ruptures.

The size distribution of starch granules from winter wheat endosperm was determined using a Malvern Mastersizer 3000 analyzer (Malvern Instruments Ltd., Malvern, UK). The trimodal size distribution of starch granules was used, i.e., the diameter of large A-granules was >10 μm, B-granules 5–10 μm and for C-granules < 5 μm.

### 2.4. Meteorological Data

The climate of Estonia was found to be slightly continental at the experimental site. Meteorological data of the post-hibernation vegetation period in 2013–2017 were collected from a meteorological station approximately 2 km from the trial site [20]. The length of the post-hibernation vegetation period depended on the snow melting time in April. The average duration of the vegetation period (air temperature permanently above 5 °C) was 175–190 days. Later snowfall in 2013 and 2017 and lower temperature values during the vegetation period in 2017 caused a later harvesting time of winter wheat. In 2015, lower temperatures and regular precipitation during the post-hibernation period resulted in record grain yields of winter wheat. The most unfavorable season for grain yield formation were the weather conditions in 2016, when the amount of precipitation in May was only 2 mm and temperature data during the post-hibernation vegetation period were higher than the long-term average. In addition, the wintering of wheat failed in 2016, where the

number of plants per unit area decreased two-fold, resulting in a grain yield 3.4 times lower than the recorded yield in 2015.

### 2.5. Statistical Analysis

A statistical analysis of the collected data was performed with the software Statistica 13 (Quest Software Inc., Aliso Viejo, CA, USA). Factorial analyses of variance (ANOVA) and two-factor ANOVA were used to test the effect of cropping systems and the experimental year on granule-size distribution and flour yield. Fisher's least significant difference test for homogenous groups was used for testing significant differences between treatments and between years. The means are presented with their standard errors (±SE). The level of statistical significance was set at $p < 0.05$, if not indicated otherwise.

## 3. Results

### 3.1. Effect of N Treatment and Weather Conditions on Starch-Gluten Interaction

The size distribution of starch granules and the amount of protein and gluten accumulated during the filling period of kernels depended on the length of the grain-filling period and availability of N from organic or mineral N fertilizers.

The size distribution of starch granules was significantly influenced by weather conditions during the post-anthesis grain-filling period of winter wheat (calculated proportion of variance for weather conditions and N treatment was 38 and 17%, respectively). The diameter of A-type starch granules was significantly smaller ($p < 0.01$) in 2013, 2016 and 2017, when the proportion of C-type starch granules in kernels was higher due to 3–5 days longer grain filling period (from flowering up to physiological maturity stage) than in other trial years (Figure 1). The diameter of the A-type starch granules was larger in trial treatments of Org II (winter cover crops and manure) and N150, where the mean diameter of granules was 22.8 ± 1.9 and 20.9 ± 3.2 μm, respectively.

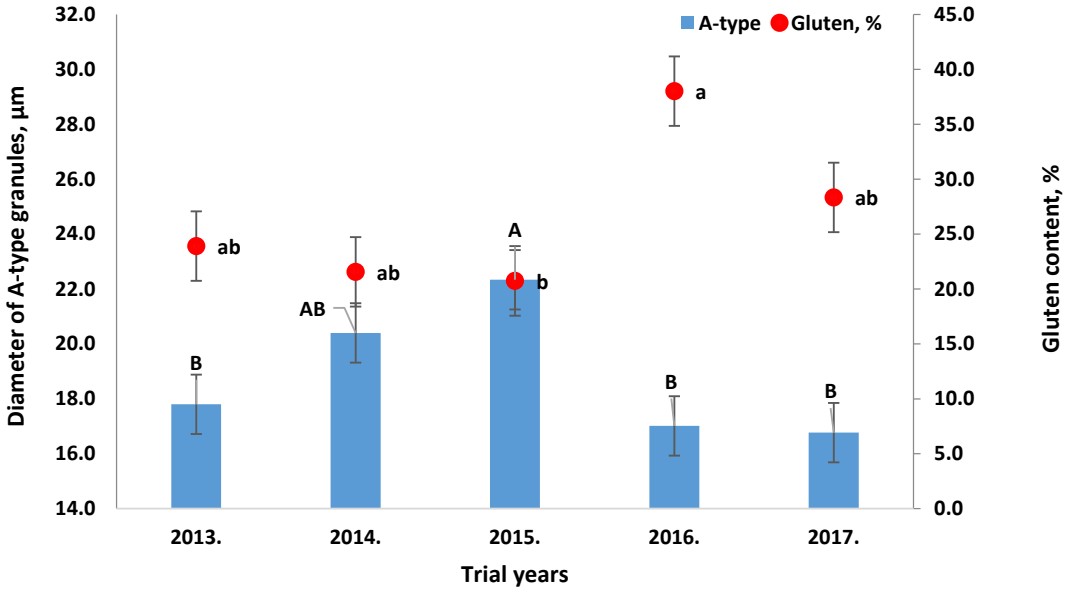

**Figure 1.** Mean diameter of A-type starch granules (μm) and gluten content (%) values over treatments during different trial years. The means marked with the same letter do not differ significantly from each other.

The gluten content of flour was most significantly influenced by the weather conditions, which in turn influenced the nitrogen availability during the grain-filling period (proportion of variance for weather conditions was 83%); the gluten content was the highest in the N150 treatment varying between 24–43% in the trial years. Higher amounts of gluten in wheat kernels were measured in the trial years with significantly smaller A-type starch

granules (Figure 1), whereas there was no significant correlation between gluten content and starch-granule diameter.

N treatments had a strong effect on the protein content (R = 0.70, *p* < 0.001; Table 1). The protein content (PC) of winter wheat did not vary significantly between organic and conventional fertilizing treatments. Mean PC was highest in conventional treatment N150, of up to 3.9 percentage points higher than mean PC values of all other treatments. The PC was influenced by the weather conditions of each growing year, while higher values were obtained in years with a lower level of grain yield; the higher mean PC values over treatments were 12.4 ± 0.30 and 14.6 ± 0.48% in 2014 and 2016, respectively, which proved to be up to 2.3 and 4.5 percentage points higher than that of other years, respectively. PC correlated positively with dough quality number (R = 0.70, *p* < 0.001).

**Table 1.** Rheological values from doughs obtained from different N treatments.

| Indicator. | Org 0 *** | Org I | Org II | N0 | N50 | N100 | N150 |
|---|---|---|---|---|---|---|---|
| PC, % ** | 11.2 ± 0.7 b | 11.5 ± 0.6 b | 11.3 ± 0.9 b | 11.3 ± 0.6 b | 11.6 ± 1.1 b | 12.9 ± 1.1ab | 13.6 ± 0.7 a |
| WGC, % | 25 ± 2.6 b * | 25 ± 2.8 ab | 25 ± 4.2 ab | 25 ± 2.8 ab | 25 ± 3.8 ab | 29 ± 3.4 ab | 31 ± 3.3 a |
| GI,% | 86 ± 2.2 a | 84 ± 2.2 a | 87 ± 1.9 a | 87 ± 2.7 a | 83 ± 4.9 a | 82 ± 3.3 a | 73 ± 3.5 b |
| WAC,% | 57 ± 0.8 b | 57 ± 0.7 b | 57 ± 0.8 b | 58 ± 0.6 b | 58 ± 0.7 b | 60 ± 0.9 a | 61 ± 0.4 a |
| DDT, min | 2.08 ± 0.2 b | 2.02 ± 0.2 b | 2.02 ± 0.2 b | 2.24 ± 0.3 ab | 2.16 ± 0.2 ab | 3.00 ± 0.3 ab | 3.54 ± 0.2 a |
| S, min | 4.20 ± 0.55 a | 4.17 ± 0.42 a | 4.07 ± 0.50 a | 4.39 ± 0.43 a | 4.28 ± 0.55 a | 5.08 ± 0.40 a | 6.20 ± 0.38 a |
| DS, FE | 67 ± 12.5 a | 65 ± 5.9 a | 68 ± 11.4 a | 67 ± 9.0 a | 66 ± 9.4 a | 56 ± 8.1 a | 44 ± 5.4 a |
| W, 10e-4 J | 204 ± 18 a | 204 ± 7 a | 213 ± 11 a | 233 ± 11 a | 217 ± 11 a | 234 ± 10 a | 255 ± 17 a |
| DQN, mm | 49 ± 11.2 a | 48 ± 7.3 a | 47 ± 9.8 a | 50 ± 8.5 a | 53 ± 9.5 a | 65 ± 9.7 a | 81 ± 7.4 a |

* Different letters denote significant difference; ** PC—protein content, WGC—wet gluten content, GI—gluten index, WAC—water absorption capacity, DDT—dough development time, S—dough stability time, DS—degree of dough softening, W—force required to create a bubble, DQN—dough quality number; *** Org0 and N0 = control treatments of organic and conventional farming, respectively; Org I = organic treatment with winter cover crops (CC); Org II = in addition to CC the cattle manure application; N50, N100 and N150 = amounts of mineral N50, 100 and 150 kg N ha$^{-1}$, respectively.

Wet gluten content (WGC) correlated strongly with PC (R = 0.77, *p* < 0.001). Significantly higher values of gluten were obtained from kernels of treatment N150 (Table 1) and in the trial years of 2016 and 2017 (mean WGC values over N treatments were 38 ± 0.7 and 28 ± 0.6%, respectively). According to the protein and gluten content values, only the flour obtained from the N150 treatment belonged to the elite class; flour from the N100 treatment belonged to A class and the flours of all other variants were of low quality [25]. The WGC correlated positively with water absorption capacity and dough development time (R = 0.57 and 0.51, respectively; *p* < 0.001 for both values).

The gluten index (GI) characterizes the strength of the dough. GI values were mostly influenced by N treatment and the second factor, i.e., weather conditions during each trial year had little or no effect. The calculated proportion of variance for treatment and weather conditions was 36 and 14%, respectively. N availability in different treatments had a strong effect on GI (R= −0.56, *p* < 0.001); lower values were measured in wheat flours obtained from treatments N100 and N150 that were fertilized twice after wintering (Table 1). The mean values of GI in organic treatments and conventional treatments with lower amounts of N ranged between 83–87%, which was up to 5 and 15% higher than that of N100 and N150 value, respectively. For forming dough with good strength, GI values between 60–95% are considered acceptable [26]. According to Cubadda et al. [27], GI may describe whether gluten quality is weak (GI < 30%), normal (GI = 30–80%), or strong (GI > 80%).

The GI values of this field trial were negatively correlated with dough-development time (R = −0.53, *p* < 0.001), water absorption capacity (R = −0.49, *p* < 0.01), dough stability (R = −0.49, *p* < 0.01) and finally with dough-quality number (R = −0.56, *p* < 0.001). The correlation between GI and diameter values of smaller starch granules was significantly negative (coefficient R for C- and B-type granules was −0.32 and −0.34, respectively). In

general, the mean value of GI across treatments and trial years was 83 ± 1.3%, whereas GI varied under different trial years by 69–96% (Figure 2).

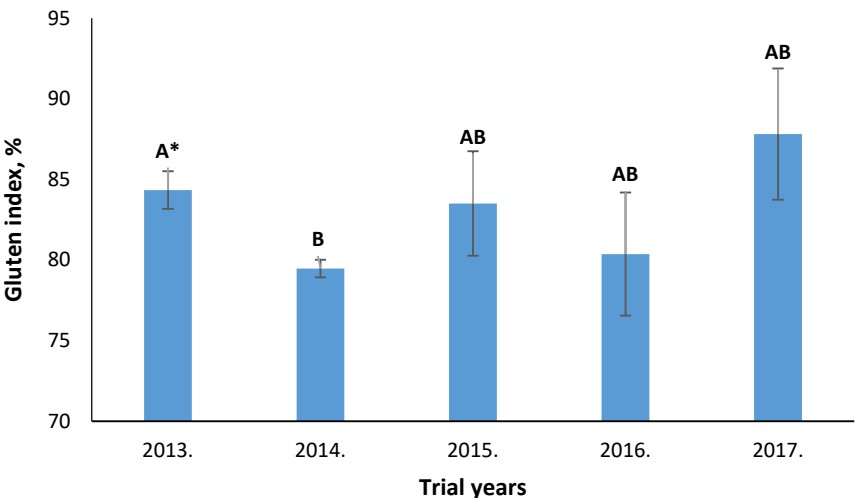

**Figure 2.** Gluten index values under different trial years. * The means marked with the same letter do not differ significantly from each other.

### 3.2. Rheological Properties of Wheat Dough

When water is mixed into wheat flour, it forms dough. The water absorption capacity (WAC) of flour is one of the indicators of the quality of dough. WAC values were influenced by both trial factors—by weather conditions in trial years and N treatments; the calculated proportion of variance was 36 and 46%, respectively. WAC of flour correlated positively with PC and WGC (R = 0.83, $p < 0.001$ and 0.57, $p < 0.001$, respectively) and positively with diameter values of small B- and C-type starch granules (R = 0.32 and 0.34, respectively). The differences in WAC values between treatments were not significant, except for the N100 and N150 treatment with a WAC of up to 61% (Table 1). The diameter of smaller starch granules was larger in 2014 and 2016 [21], and WGC was higher in 2016 and 2017 (Figure 1); mean values of WAC ranged between 58–60% in 2014, 2016 and 2017, which was up to 3 percentage points higher than that of other trial years. Wheat flours with WAC values greater than 58% were classified as strong flours [28].

Of the factors studied, the dough development time (DDT) was only significantly influenced by N treatment (R = 0.52, $p < 0.001$) and it correlated positively with PC and WGC (R = 0.83, $p < 0.001$; R = 0.51, $p < 0.001$, respectively). DDT of treatments ranged between 1.46–4.09 min, whereas only the N150 treatment had a mean value of DDT significantly longer than that of organic treatments, of up to 2.23 min longer (Table 1). According to Kassomeh [28], DDT may describe whether dough is weak (DDT is less than 2.5 min), medium (2.5–4.0 min) or strong (4.0–8.0 min). DDT was positively correlated with dough quality number (R = 0.86, $p < 0.001$).

The stability of dough (S) was most affected by PC (R = 0.48, $p < 0.01$). The correlation analysis also demonstrated that dough stability was better when the diameter of smaller starch granules (C- and B-type) was larger (R = 0.32). The most stable dough was made from N150 flour. The mean values of dough stability over trial years did not differ significantly between any of the treatments (Table 1), varying between 3.33–6.58 min. This particular time of S is classified as the medium stability [28,29].

The degree of dough softening (DS) 10 min after initiating mixing and also 12 min after attaining the maximal peak was significantly influenced by both trial factors (calculated proportion of variance for trial year and N treatment was 63 and 17%, respectively). The softening degree was significantly low in 2014 (46 ± 3.9%), followed by 2013 and 2016 (up to 10% higher than that of 2014). Mean DS values (10 min after beginning) of N treatments as an average of trial years ranged between 39–76 FE, which is classified as

medium [28]. Lower softening degree values were measured in the doughs derived from flour of conventional treatments of N100 and N150. As the differences between trial years were significant, the mean effect of treatments over trial years was found to be non-significant (Table 1). The coefficient of variation was higher for organic treatments (ranged between 20–42%; for conventional treatments 27–32% as represented in Table 2).

**Table 2.** Coefficient of variation (%) of different dough quality parameters over trial years (2013–2017) for all trial treatments.

| Indicator | Org 0 ** | Org I | Org II | N0 | N50 | N100 | N150 |
|---|---|---|---|---|---|---|---|
| WAC, % * | 3.2 | 2.6 | 3.2 | 2.2 | 2.6 | 3.3 | 1.5 |
| DDT, min | 40.1 | 30.7 | 42.5 | 44.6 | 36.7 | 41.3 | 15.6 |
| S, min | 47.6 | 36.8 | 45.9 | 34.7 | 46.1 | 29.5 | 22.4 |
| DS, FE | 41.9 | 20.2 | 37.4 | 29.9 | 31.8 | 32.5 | 27.4 |
| FQN, mm | 51.1 | 33.9 | 47.1 | 37.9 | 40.2 | 33.4 | 20.4 |
| AVERAGE | 33.0 | 22.8 | 32.2 | 26.8 | 29.8 | 26.6 | 16.6 |

* WAC—water absorption capacity, DDT—dough development time, S—dough stability time, DS—degree of dough softening, PC—protein content, DQN—dough quality number; ** See explanation under Table 1.

DS values were positively correlated with GI (R = 54, $p < 0.001$), negatively with grain PC ($R_{10min} = -0.55$, $p < 0.001$) and with diameter values of the smallest C-type starch granules (R = −0.32). The larger granules (A and B types) did not significantly affect the softening degree of dough.

According to the alveograph results, the force required to create a bubble (W) was mostly affected by N treatment (R = 0.50, $p < 0.001$), followed by the weather conditions of the trial year (R = 0.38; Table 1). The values of force required for bubbles ranged, in organic and conventional systems, between 204–213 and $217–255 \times 10^{-4}$ J, respectively, while differences between all treatments were not significant. Lower values of this force were obtained in years with a higher level of grain yield and lower content of protein (in 2015 and 2017).

The dough quality number (DQN) was influenced by both trial factors, whereas the annual weather conditions had a much stronger effect than N treatment (calculated proportion of variance was 46 and 29%, respectively). Improved values of DQN were obtained in 2014, in which the quality values ranged between 64–108 mm, followed by 2016, when quality values ranged from 59–77 mm. The lowest values of dough quality were obtained in 2017, when quality values ranged in all treatments, except for N150, between 21–31 mm (Figure 3). DQN depended significantly on the availability of nitrogen during the grain-filling period, and it correlated positively with kernels PC (R = 0.70, $p < 0.001$; Table 1). The mean dough quality values of the treatments over the trial years did not differ significantly because of their great variability.

The final DQN value was found to be the result of an interaction of several factors and Table 2 shows the correlation coefficients between the DQN and the indicators that influenced its value more significantly. In particular, the PC and WGC of the flours were decisive in the formation of the DQN value, as these parameters greatly influenced the properties of the dough. B- and C-type starch granules indirectly affected the values of DQN in the interaction with WAC and GI (Table 3).

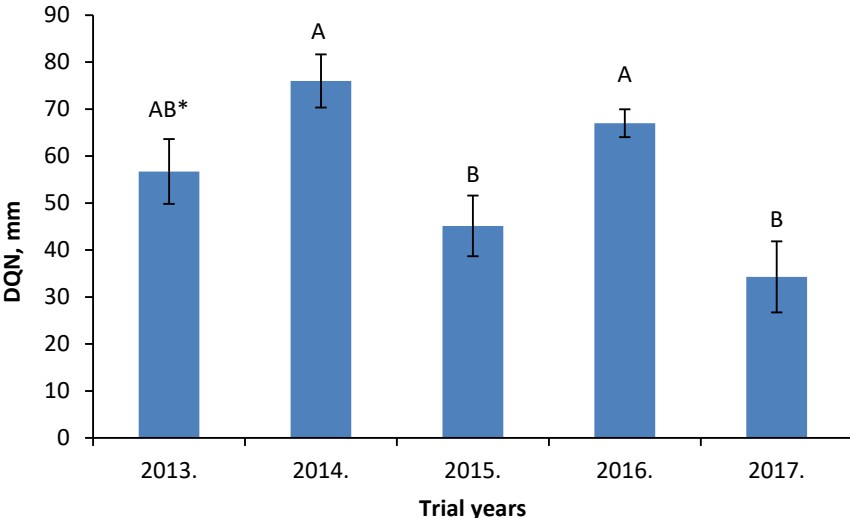

**Figure 3.** Mean values of dough quality number (DQN; mm) of winter wheat Fredis over N treatments. * Different letters denote significant difference.

**Table 3.** Factors that most affected value of dough quality number (DQN) expressed as a correlation coefficient (R).

| Factors | R * | p ** |
|---|---|---|
| 1. Dough stability (S) | 0.94 | $p < 0.001$ |
| 2. Degree of dough softening (DS) | −0.89 | $p < 0.001$ |
| 3. Dough development time (DDT) | 0.86 | $p < 0.001$ |
| 4. Water absorption capacity (WAC) | 0.73 | $p < 0.001$ |
| 5. Grain protein content (PC) | 0.70 | $p < 0.001$ |
| 6. Gluten index (GI) | −0.56 | $p < 0.001$ |
| 7. WAC x diameter of B- and C-type starch granules | 0.33 | $p < 0.05$ |
| 8. GI x diameter of B- and C-type starch granules | −0.32 | $p < 0.05$ |

* Correlation coefficient; ** *p* value indicates significance at 95, 99 and 99,9%.

### 3.3. Variation of Dough Quality Number

The results of this field trial demonstrate that dough rheological properties and its quality were significantly influenced by both trial factors; the proportion of variation for year and treatment were 46 and 29%, respectively. Both of these factors in turn influenced the accumulation of protein and gluten in grains of winter wheat and the size distribution of starch granules. Due to the significant variation in values of dough rheological properties and quality over the years, there were often no significant differences found between the treatments. The coefficient of variation for different dough properties (Table 3) indicated that the smallest variability between these values was for the N150 treatment, where WGC was the highest and the GI was the lowest (Table 1).

### 4. Discussion

When wheat flour is mixed with water, natural hydrocolloid gluten interacts with water molecules to create wheat dough, embracing a three-dimensional network, filled with starch granules [18]. Gluten traps the gases generated during the fermentation process, while starch significantly affects the rheological properties of the dough. The specific interaction between gluten and starch in the microstructure of wheat dough has been shown to influence dough behaviour [30]. Therefore, determining the properties of separated gluten and starch independently may not fully explain the behaviour of wheat dough [18]. The topic of this article is the study of the effect of mineral and organic nitrogen fertilizers on the gluten content of wheat dough, the combined effect of starch granule size and gluten on the rheological properties of the dough, and the effect of weather conditions on the stability of dough quality.

The first hypothesis was that wheat grown under a mineral nitrogen background has dough of a higher quality, which was mostly confirmed by our results. The quality of the dough was assessed through its rheological properties and the results showed that quality depended on a combination of several factors, but in particular on the quantity and quality of protein and gluten, which in turn depended on both of the studied trial factors (weather conditions in trial years and N treatment, i.e., mineral or organic N availability during grain filling period). The protein and gluten content and therefore, the wheat-dough quality, were better in 2014 and 2016, when the grain yield level was lower [20], which is also in line with the findings of Giancaspro et al. [31]. The mean protein and gluten content over the trial years were highest in flours obtained from N treatment N150.

The quality of dough also depended on N availability as well as on the size distribution of the starch granules formed during the grain filling period.

The water absorption capacity (WAC) is one of the indicators of higher dough quality in flours with a smaller diameter of starch granules [4,18]. Previous studies found that smaller granules have a higher surface-to-volume ratio and are able to hydrate and swell more efficiently and bind more water than larger granules [13,32]. Therefore, an increased content of small granules should increase the flour water absorption capacity [33,34]. In our experiment, the WAC values also correlated positively with B- and C-type starch granule diameters, whereas the highest values of WAC were obtained when the diameter of smaller granules ranged between 3.2–9.0 μm, i.e., the WAC was higher when the diameter of smaller starch granules was larger (the diameter of the measured C- and B-type starch granules ranged between 2.45–9.84 μm). The WAC values were much better in treatments N100 and N150, which was caused mostly by higher contents of protein and gluten (Table 1). In this field experiment, the calculated proportion of variance showed that N treatment had a stronger effect on WAC values than the weather conditions.

The development time and stability of dough in this trial were longer and the degree of dough softening was lower in treatments of N100 and N150 due to a higher PC and WGC; the stability time of dough was ca 2 min longer in these treatments than that of other treatments. These results are consistent with those of Gao et al. [18], because the micro-structure of dough with higher PC and WGC contains fewer gaps between gluten and starch. The dough for which PC and WGC are fully integrated with each other in their sizes was found to be more stable. Gao et al. [18] found that wheat flours with low PC and WGC may contain irregular and larger holes. Zi et al. [35], discussing the starch granule effects on the stability of dough, found that large starch granules or fewer small starch granules are both associated with weak dough processing properties because of heterogeneity of the gaps and holes in the gluten network. Cao et al. [4] reported that starch granules act as the filling particles in a protein–starch matrix. In this experiment, the dough stability due to WAC was lower in organic and N0 and N50 treatments, probably due to irregular and larger holes in the gluten network with a lower filling degree and less interaction between small starch granules and gluten; such dough is unstable. According to Kulhomäki and Salovaara [29] the dough stability time of 4–12 min indicates a good quality of dough. In our experiment, the dough stability times in the organic treatments and in the N0 and N50 treatments were close to the lower limit, probably because of the irregular and insufficient accumulation of nitrogen in grains during the grain filling periods of different trial years, which resulted in a lower protein and gluten content. The unstable availability of N in these variants also caused great variability in the quality of the dough (Table 3).

The gluten index (GI) characterizes the ratio of gliadins to glutelins. GI, dough development time and stability are all critical parameters reflecting dough strength, and higher values of these parameters indicate stronger dough [36,37]. It has been suggested that gliadins generally contribute to dough viscosity and glutelins contribute to dough elasticity [38,39]. Oikonomou et al. [40] found that GI values are positively correlated with protein content and the level of fertilization with nitrogen. In our experiment, we found the opposite to be true—GI values were negatively correlated with protein content and amounts of nitrogen fertilizer applied, which is in line with the results reported by

Borkowski et al. [26]; higher values of GI were found to be associated with lower values of protein in most treatments (over 80% and 11.2–11.6%, respectively), except N150, where the mean value of GI was 73 ± 3.5% and PC was up to 2.4% higher than that of other treatments. GI values over 80% indicate an increased proportion of glutenins, i.e., increased resistance to extension of dough [36]. Too strong wheat dough does not rise well and does not produce an airy loaf. Due to higher PC and WGC the mean values of GI were up to 9% lower in 2014 and 2016 than those of other years.

Dough quality was influenced by several factors, with the nitrogen fertilization regime having the greatest impact. In our experiment, the dough with a higher DQN was obtained from flours with lower values of GI (treatment N150, GI = 73%) and with a medium diameter of B- and C-type starch granules. At a constant protein content and optimal glutenin-to-gliadin ratio, the dough stability time was found to be longer, the degree of softening lower and dough quality higher.

The second hypothesis posited that the quality of the dough would be more variable in organic treatments. The results of this trial showed that the amount of mineral N 150 kg ha$^{-1}$, applied in two parts before flowering, ensured stable plant growth and protein and gluten content in different years. Conventional treatments with lower mineral N amounts and organic treatments (fertilizing with manure or using winter cover crops as catch crops) were more sensitive to variable weather conditions than N150. The amount of organic N available to plants was more significantly affected by the distribution of precipitation and temperatures during the growing season, which resulted in the greater variability of values over years; the mean variability of different dough properties over trial years was observed in the organic treatment trials (Org 0, Org I and Org II) at 1.4–2.0 times higher than that of the N150 treatment. A high variability in weather conditions during this study period (2013–2017 years) caused the higher value for the proportion of variation for the year; this resulted in variable values of winter wheat grain yield, protein and gluten content (see more detail in [20]) and therefore, in variable dough rheological properties. Oikonomou et al. [40] reported that higher values of the coefficient of variation indicate stronger relationships between treatments and weather conditions. The high variability in the protein content of the N50 treatment indicates that this specific amount of mineral N was not always sufficient for a full or optimal accumulation of protein during the grain-filling period across all trial years.

## 5. Conclusions

Taken together, these analytical results of winter wheat dough strongly indicate that, of the seven treatments tested, mineral fertilization with the highest nitrogen rate (150 kg/ha) is crucial for achieving high quality dough. On the other hand, organic cropping systems, even with added manure, are still more vulnerable to the vagaries of the air–soil microclimate, and this sensitivity is reflected in dough quality. Future studies on different crop rotations and different organic fertilizers should focus on soil nutrient density, to investigate ways in which the soil/root uptake mechanisms can be made more harmonious for plant growth and the plant's own holistic health as reflected in its protein–starch interplay. Furthermore, in the context of increasing organic production, it is important to scrutinize other solutions in the value chain of bread-making that can account for the variability in grain quality and can improve wheat-dough quality.

**Author Contributions:** Conceptualization, I.K.; methodology, I.K. and M.A.; resources, E.L.; writing—original draft preparation, I.K. and M.A.; writing—review and editing, I.A., E.L., R.K. and T.T.; visualization, M.A. and I.K. All authors have read and agreed to the published version of the manuscript.

**Funding:** This research was funded by the Estonian Research Council grant PRG 1260 and supported by ERDF and Estonian Research Council via project RESTA28. The study has also been supported by Estonian University of Life Sciences projects 8–2/T13001PKTM, P170261PKTT, P170062PKTM and by ERA–NET CORE-ORGANIC II project TILMAN—ORG. I.A. was partly supported by Proteins Easy Corp.

**Institutional Review Board Statement:** Not applicable.

**Informed Consent Statement:** Not applicable.

**Data Availability Statement:** Not applicable.

**Acknowledgments:** Many thanks to the crop science team, especially to Enn Lauringson, Are Selge, Liina Talgre and Vyacheslav Eremeev who have contributed towards the experiment initation and management. The technical assistance of Rõhu experimental station from the Estonian University of Life Sciences is gratefully acknowledged.

**Conflicts of Interest:** The authors declare no conflict of interest. The funders had no role in the design of the study; in the collection, analyses, or interpretation of data; in the writing of the manuscript, or in the decision to publish the results.

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
