# Peer review of "The Combined Effect of Nitrogen Treatment and Weather Conditions on Wheat Protein-Starch Interaction and Dough Quality"

_agriculture, doi:10.3390/agriculture11121232_

Round 1

Reviewer 1 Report

Comments

This manuscript reported a interesting topic that organic and mineral nitrogen fertilizers on the gluten-starch interaction and rheological properties of winter wheat dough. Although this manuscript looks now better, some revisions are still needed for this manuscript. Some corrections, clarifications and language adjustments are required.

Special comments

Title

The title does not match the research content.

Abstract

I understood our results in the abstract, you should consider these sentences in lines 13 to 18. The experimental design contained two factors, you should think more.

Introduction

The (1) and (2) are better in the lines 61-61.

Materials and Methods

You should explain more about the fertilizer. And I also noticed that this experimental field planted many crops, you only choose the winter wheat for this experiment. Is this a continuous  or rotation system? You should supply some information.

You use a two factor ANOVA in lines 145-146, you should detail each factor here.

Results

Lines 152-155 This paragraph was the aim of this present manuscript.

The y line in figure is missing.

The effects of different N fertilizers was significant, while little information is in lines 180-183.

Did you compare the effects of different N fertilizers and trial years.

Author Response

We are thankful to the reviewer for the helpful comments. Please see the attachment for the point-by-point responses. 

Reviewer 2 Report

I have reviewed this manuscript with a good deal of interest. The manuscript's data is quite interesting and phenomenal. The authors have described the methods in a comprehensive way.

here are my comments.

the abstract is not really well written. It should contain the results in a quantitative manner. Aslo please use the standard style of units e.g., line 13.

while the authors have provided sufficient information in the introduction, they failed to describe comprehensively the research problem and formulation of their scientific question. There should also be a gap statement.

methods section is quite good.How the authors checke the data normality and homogeneity of the variance?

results can be modified in a bit expressive way. The lack of focus is crucial.

lines 152-155, redundant and can be deleted.

Discssion section need some improvements in justifying the results obtained based on other related researches. the mechanisms of discussion should be explored further.

The conclusion section should be based on the findings obtained. I suggest deleting general statements here.

Author Response

(The authors gave the same response as above.)
